# Production of Minor Ginsenosides from *Panax notoginseng* Flowers by *Cladosporium xylophilum*

**DOI:** 10.3390/molecules27196615

**Published:** 2022-10-05

**Authors:** Yin-Fei Li, Ying-Zhong Liang, Xiu-Ming Cui, Lin-Jiao Shao, Deng-Ji Lou, Xiao-Yan Yang

**Affiliations:** 1Faculty of Life Science and Technology, Kunming University of Science and Technology, Kunming 650500, China; 2School of Chemical, Biological and Environmental Sciences, Yuxi Normal University, Yuxi 653100, China; 3Yunnan Key Laboratory of Sustainable Utilization of Panax Notoginseng, Kunming 650500, China

**Keywords:** *Panax notoginseng* flowers, *Cladosporium xylophilum*, transformation, minor ginsenosides, enzyme preparation

## Abstract

*Panax notoginseng* flowers have the highest content of saponins compared to the other parts of *Panax notoginseng*, but minor ginsenosides have higher pharmacological activity than the main natural ginsenosides. Therefore, this study focused on the transformation of the main ginsenosides in *Panax notoginseng* flowers to minor ginsenosides using the fungus of *Cladosporium xylophilum* isolated from soil. The main ginsenosides Rb_1_, Rb_2_, Rb_3_, and Rc and the notoginsenoside Fa in *Panax notoginseng* flowers were transformed into the ginsenosides F_2_ and Rd_2_, the notoginsenosides Fd and Fe, and the ginsenoside R_7_; the conversion rates were 100, 100, 100, 88.5, and 100%, respectively. The transformation products were studied by TLC, HPLC, and MS analyses, and the biotransformation pathways of the major ginsenosides were proposed. In addition, the purified enzyme of the fungus was prepared with the molecular weight of 66.4 kDa. The transformation of the monomer ginsenosides by the crude enzyme is consistent with that by the fungus. Additionally, three saponins were isolated from the transformation products and identified as the ginsenoside Rd_2_ and the notoginsenosides Fe and Fd by NMR and MS analyses. This study provided a unique and powerful microbial strain for efficiently transformating major ginsenosides in *P. notoginseng* flowers to minor ginsenosides, which will help raise the functional and economic value of the *P. notoginseng* flower.

## 1. Introduction

*Panax notoginseng* (Burk.) F.H. Chen (Araliaceae) is a traditional precious Chinese herbal medicine, which mainly grows in the Yunnan and Guangxi provinces in southwest China. Saponins are the main bioactive ingredients in different parts of *P. notoginseng*. The part of the *P. notoginseng* flower (PNF) contains more than 20% of the total saponins, which is the highest saponin content in the whole plant [1,2,3]. The ginsenosides Rb_1_, Rb_2_, Rb_3_, and Rc and the notoginsenosides Fa and Fc are the major saponins in the PNF and belong to the protopanaxadiol (PPD) ginsenosides, but minor ginsenosides with minimal levels have higher pharmacological activity than the major natural ginsenosides. Studies have shown that minor saponins containing less sugar may show higher bioavailability, better cell permeability, and other advantages; so, minor saponins show higher pharmacological activity [4,5,6]. For example, ginsenoside Rd_2_ can prevent or treat thrombotic diseases; notoginsenoside Fe can treat cardiovascular and cerebrovascular diseases and inhibit diet-induced obesity [7,8]. The minor ginsenosides have similar structures to the major ginsenosides and can be transformed from the major ginsenosides. Therefore, we can prepare minor ginsenosides from major ginsenosides in PNF. At present, the main methods of obtaining minor ginsenosides include physical transformation, chemical transformation, biological transformation and cloned ginsenoside enzyme transformation [9]. The biotransformation method uses one or more enzymes produced by microorganisms under suitable conditions to modify the ginsenosides by hydrolysis, hydration, and dehydration reactions to obtain high pharmacological activity of the ginsenosides or new saponin derivatives and can be used as an auxiliary means to study the mechanism of drug metabolism [10], such as that of ginsenoside Rg_3_ with antitumor and anti-inflammatory effects from ginsenoside Rb_1_ transformed by the endophytic bacterium *Burkholderia* sp. GE 17-7 isolated from *Panax ginseng* [11,12] and ginsenosides C-K with anti-cancer and anti-inflammatory effects from ginsenoside Rb_1_ transformed by *Aspergillus niger* XD101 isolated from the soil of *Panax notoginseng* [13]. Compared with other transformation methods, the biological transformation method has the advantages of mild reaction conditions, stable product, easy separation, little environmental pollution, efficient transformation, and fewer by-products [14,15]. Ginsenosidase for the biotransformation of saponins can be divided into four types: ginsenosidase type-I can hydrolyze the PPD ginsenosides C-3 and C-20 glycoside bonds; ginsenosidase type-II can hydrolyze the PPD ginsenoside C-20 glycoside bond; ginsenosidase type-III can hydrolyze the PPD ginsenoside C-3 glycoside bond; and ginsenosidase type-IV can hydrolyze the PPT-type ginsenosides C-6 and C-20 glycoside bonds [13,15,16].

In our study, many fungi were obtained from the soil. The strains were screened by the transforming activity of ginsenoside Rb_1_. We hope to find a strain with a high conversion rate of the substrates and a high pharmacological activity of the transformed product. We reported for the first time a strain with a high transformation activity that could transform the major saponins in PNF, such as the ginsenosides Rb_1_, Rb_2_, Rb_3_, and Rc and the notoginsenoside Fa, into the ginsenosides F_2_ and Rd_2_, the notoginsenosides Fd and Fe, and the ginsenoside R_7_, respectively, with minimal by-products. The transformation products were studied by TLC, HPLC, and MS analyses, and the biotransformation pathways of the major ginsenosides were proposed. In addition, the purified enzyme of the fungus was prepared with the molecular weight of 66.4 kDa. Most of the enzymes produced by the active strains in this study were type-I ginsenosidase, which mainly hydrolyzed the lateral glucose at the C-3 and C-20 positions. Additionally, three monomer ginsenosides (ginsenoside Rd_2_ and notoginsenosides Fe and Fd) were isolated and elucidated from the transformation products.

## 2. Results and Discussion

### 2.1. Ginsenoside-Transforming Activity Screening and Characterization of Strain S7

PPD ginsenosides are the main components of PNF; among them, ginsenoside Rb_1_ is one of the major saponins in PNF and a representative of the protopanaxadiol (PPD) ginsenosides.

We screened six strains for ginsenoside Rb_1_ transformation activity by the TLC methods. The results showed that the strains S7, S3, and S17 have less polar spots on the TLC, which indicated that the three strains have the ability to transform ginsenoside Rb_1_ into another saponin. Compared with strains S3 and S17, strain S7 has a higher transformation rate; there was almost no spot of substrate on the TLC, indicating that the transformation substrate was almost exhausted. In addition to this, the main product of ginsenoside Rb_1_ by the strain S3 and S17 was ginsenoside Rd, which is the main component of the flower, not the target rare saponin, while the transformation product of strain S7 was the rare ginsenoside, with no intermediate product. The TLC analysis of the transformation products by different strains showed that strain S7 exhibited a significant ability to transform Rb_1_ compared to the other stains (Appendix A). So, strain S7 was selected for the further experiments.

After strain S7 was cultured on PDA medium for 4 days, the following colony characteristics were observed: the surface was olive green and villous and the colony was flat, as shown in Appendix A. Its morphological characteristics were observed under light microscope as follows: the conidiophores were erect, slightly curved, nodal, septate and slightly branched. The side formed a conidia chain which was branching and light brown. The conidia morphology was variable and smooth, nearly spherical, elliptic, and long cylindrical, as shown in Appendix A [17,18]. Based on the sequencing of the ITS rDNA gene and a comparison in the GenBank database, it was found that strain S7 belonged to the genus *Cladosporium* and exhibited significant similarity to *Cladosporium xylophilum* in Appendix A.

### 2.2. Qualitative and Quantitative Analysis of Major Saponins in PNF by HPLC

Using 8 mg PNF extract (marked as m, m = 8 mg), they were dissolved in 1 mL methanol (marked as V_t_, V_t_ = 1 mL) as the analysis sample. The injection volume was 20 µL. The purpose of the HPLC analysis is to obtain the peak area of each saponin and calculate the contents of each saponin of the major saponins in PNF according to the standard curve (marked as m_1_).
m_1_ = C × V_i_ (V_t_/V_i_)(1)
C: the concentration obtained by plugging the peak area of the major saponins into the standard curve, mg/mL; Vi: the injection volume, 20 µL.
(2)$$\mathrm{Content}\;\left(\%\right)=\frac{m_{1}}{m}\times 100\%$$

The purpose of analyzing the major saponins in the PNF is to calculate the conversion rate of those saponins during the biotransformation process by *C. xylophilum.* The qualitative and quantitative analyses of the major saponins in the PNF by the HPLC method are shown in Figure 1. The results showed that the contents of the major saponins (notoginsenosides Fa and Fc and ginsenosides Rb_1_, Rb_2_, Rb_3_, Rd, and Rc) in the PNF were 2.80, 0.29, 0.60, 0.52, 4.80, 0.15, and 2.40%, respectively.

### 2.3. HPLC Analysis the Dynamic Change of Major Saponins in PNF Transformed by C. xylophilum

During the biotransformation process of the major saponin in the PNF by *C. xylophilum*, it was regularly monitored by HPLC analysis (Figure 2). As shown in Figure 3, the notoginsenoside Fa and the ginsenosides Rb_1_, Rb_2_, Rc, Rb_3_, and Rd that comprised the major portion of the PNF were rapidly transformed into other saponins in the early stage of the reaction (1–5 days). After 10 days of reaction, the notoginsenoside Fa and the ginsenosides Rb_1_, Rb_2_, Rb_3_, and Rd were completely transformed by *C. xylophilum*, and the conversion rate reached 100%. After 15 days of reaction, only notoginsenoside Fc and ginsenoside Rc were left in the PNF, and the final conversion rates were 53.4 and 88.5%, respectively.
(3)$$\mathrm{Conversion}\;\mathrm{rate}\;\left(\%\right)=\frac{\mathrm{total}\;\mathrm{saponins}\;-\;\mathrm{remaining}\;\mathrm{saponins}}{\mathrm{total}\;\mathrm{saponins}}\times 100\%$$

### 2.4. HPLC Analysis of the Transformation Pathways of Monomer Ginsenosides Rb_1_, Rb_2_, Rb_3_, Rc, Notoginsenosides Fa and Fc by C. xylophilum

In order to further verify the transformation pathways of the main saponins in the PNF, the ginsenosides Rb_1_, Rb_2_, Rb_3_, and Rc and the notoginsenosides Fa and Fc were used as substrates for the transformation experiments, respectively.

The transformation pathway of ginsenoside Rb_1_ is proposed in Figure 4A. The ginsenoside Rb_1_ molecule contains four *β*-glucopyranosyl moieties at the C-3 and C-20 position of aglycone. Based on the results obtained by HPLC analysis (Appendix A), we can see that there are peaks of small polar products in the product, which were identified as F_2_ by comparing their retention time with the standard ginsenoside F_2_; so, we suggest that Rb_1_ was biotransformed into F_2_ by *C. xylophilum*. The biotransformation of Rb_1_ into F_2_ can occur through pathways of two types, depending on their structures. Firstly, the enzyme from *C. xylophilum* attacked the outer *β*-(1→2)-glucosidic linkage to the C-3 position of aglycone to produce Gyp17 from Rb_1_ and was then followed by the hydrolysis of the outer *β*-(1→6)-glucosidic to the C-20 position to produce F_2_ from Gyp17. Secondly, the enzyme from *C. xylophilum* attacked the outer *β*-(1→6)-glucosidic linkage to the C-20 position of aglycone to produce Rd from Rb_1_ and was then followed by the hydrolysis of the outer *β*-(1→2)-glucosidic to the C-3 position to produce F_2_ from Rd.

The transformation pathway of notoginsenoside Fa is proposed in Figure 4B. The notoginsenoside Fa contains one *α*-(1→2)-xylopyranosyl (outer) and two *β*-glucopyranosyl moieties (inner) at the C-3 position, with two *β*-glucopyranosyl moieties at the C-20 position of aglycone. Based on the results obtained by the HPLC analysis (Appendix A), we can see that there is a main peak of small polar products in the HPLC spectrum; so, we suggest that Fa can be transformed into another ginsenoside by *C. xylophilum*. The product’s molecular formula of C_53_H_90_O_22_ was determined by HR-ESI-MS at m/z 1077.5843 [M-H]- (calcd. for 1077.5845). Ginsenoside R_7_ has the same molecular formula of C_53_H_90_O_22_. Due to the existence of isomers, we analyzed the possible compounds with the same molecular formula in *Panax plants*. According to the characteristics of the saponin transformation pathway (which usually hydrolyzes one or more glycosyl fragments), we determined that the product of substrate (ginsenoside Fa) transformed by *C. xylophilum* was ginsenoside R_7_.

The enzyme from *C. xylophilum* attacked the outer *β*-(1→6)-glucosidic linkage to the C-20 position of aglycone to produce R_7_ from Fa. In addition to the HPLC analysis, the HR-ESI-MS analysis of the transformation product of notoginsenoside Fa was further verification that the product was ginsenoside R_7_, as shown in Appendix A.

The transformation pathway of ginsenoside Rb_2_ is proposed in Figure 4**C**. The ginsenoside Rb_2_ molecule contains one *α*-(1→6)-arabinopyranosyl (outer) and one *β*-glucopyranosyl moiety (inner) at the C-20 position, with two *β*-glucopyranosyl moieties at the C-3 position of aglycone. Based on the results obtained by the HPLC analysis (Appendix A), we can see that there are peaks of small polar products in the product, which were identified as Rd_2_ by comparing their retention times with standard ginsenoside Rd_2_; so, we suggested that Rb_2_ was transformed into ginsenoside Rd_2_ by *C. xylophilum*. The enzyme from *C. xylophilum* attacked the outer *β*-(1→2)-glucosidic linkage to the C-3 position of aglycone to produce Rd_2_ from Rb_2_. Similarly, the enzyme from *C. xylophilum* attacked the outer *β*-(1→2)-glucosidic linkage to the C-3 position of aglycone to produce Fe from Rc (Figure 4D and Appendix A). The enzyme from *C. xylophilum* attacked the outer *β*-(1→2)-glucosidic linkage to the C-3 position of aglycone to produce Fd from Rb_3_ (Figure 4E and Appendix A).

The enzyme of *C. xylophilum* can hydrolyze lateral glucose at the C-20 and C-3 of ginsenoside Rb_1_ to F_2_ through two pathways. In addition to this, the enzyme can hydrolyze lateral glucose at the C-20 or C-3 of notoginsenoside Fa and the ginsenosides Rb_2_, Rb_3_, and Rc through a single pathway, but cannot hydrolyze the arabinose, xylose, and inside glucose. It indicated that the enzyme from this strain was highly specific, and it could transform different saponins into specific ginsenosides.

The maximal concentration of minor ginsenosides in the transformation products of the major saponins in the PNF by using *C. xylophilum* occurred on the 10th day, as is shown in Figure 5. The contents of the minor ginsenosides F_2_ and Rd_2_ and the notoginsenosides Fd and Fe were 0.99, 0.67, 0.24, and 0.24 mg/mL, respectively.

### 2.5. Enzyme Purification and Characterization from C. xylophilum

The results of the SDS-PAGE showed that the purified enzyme was a single band, and its molecular weight was estimated to be 66.4 kDa according to the relative migration distance of the molecular weight markers in electrophoresis (Appendix A). The molecular weight of the protein was similar to that reported in the literature [19,20,21]. The *β*-glucosidase activity from *C. xylophilum* is 129 U/mL for pNP-*β*-*D*-glucopyranoside (as a dry weight base).

### 2.6. Characterization of the Crude Enzymes for Monomer Saponins Transformation

The results of the crude enzyme transformation were consistent with those of *C. xylophilum* (Appendix A). The biotransformation of the monomer saponins by the crude enzymes was studied in the pH range of 4 to 8 and the temperature range of 30 to 70 °C (Figure 6). The optimal pH for the transformation of the ginsenosides was in the range of 5–6. These results suggest that the biotransformation of ginsenosides by crude enzymes was more desirable in weak acidic conditions (pH 5–6) rather than in neutral and basic conditions. The optimal temperature was 50 °C for the biotransformation of the ginsenosides by crude enzymes.

### 2.7. Preparation and Separation of Notoginsenoside Fe, Ginsenoside Rd_2_, and Notoginsenoside Fd from Main Saponins in PNF Transformed by C. xylophilum

Compounds **1**–**3** were identified as notoginsenoside Fe (CMc_1_), ginsenoside Rd_2_ (C-O), and notoginsenoside Fd (CMx_1_) by MS and NMR analysis.

Compound **1**: notoginsenoside Fe (CMc_1_), white amorphous powder. They were determined as two *β*-linked sugars (*D*-glucopyranosy) and one *α*-linked sugar (*L*-arabinofuranosyl) by the coupling constants of the anomeric protons [δ_H_ 4.95 (1H, d, *J* = 7.5 Hz, 3-*O*-glc-1’), 4.92 (1H, d, *J* = 7.5 Hz, 20-*O*-ara-1’’’), and 5.15 (1H, d, *J* = 7.5 Hz, 20-*O*-glc-1’’)] in the ^1^H NMR spectrum. Its molecular formula of C_47_H_80_O_17_ was determined by the HR-ESI-MS at *m/z* 951.5078 [M + Cl]^−^ (Appendix A). The compound showed identical NMR signals (Appendix A) to those described in the literature [22,23].

Compound **2**: ginsenoside Rd_2_(C-O), white amorphous powder. They were determined as two *β*-linked sugars (*D*-glucopyranosy) and one *α*-linked sugar (*L*-arabinopyranosyl) by the coupling constants of the anomeric protons [*δ*_H_ 5.20 (1H, d, *J* = 7.7 Hz, 3-*O*-glc-1’), 4.98 (1H, d, *J* = 7.0 Hz, 20-*O*-ara-1’’’), and 4.92 (1H, d, *J* = 7.5 Hz, 20-*O*-glc-1’’)] in the ^1^H NMR spectrum. Its molecular formula of C_47_H_80_O_17_ was determined by the HR-ESI-MS at *m/z* 951.5088 [M + Cl]^−^ (Appendix A). The compound showed identical NMR signals (Appendix A) to those described in the literature [24,25].

Compound **3**: notoginsenoside Fd (CMx_1_), white amorphous powder. They were determined as two *β*-linked sugars (D-glucopyranosy) and one *α*-linked sugar (*L*-xylopyranosyl) by the coupling constants of the anomeric protons [δ_H_ 4.95 (1H, d, *J* = 7.5 Hz, 3-*O*-glc-1’), 4.99 (1H, d, *J* = 7.5 Hz, 20-*O*-xyl-1’’’), and 5.13 (1H, d, *J* = 7.5 Hz, 20-*O*-glc-1’’)] in the ^1^H NMR spectrum. Its molecular formula was determined as C_47_H_80_O_17_ based on HR-ESI-MS at *m/z* 961.5369 [M + HCOO]^—^ (Appendix A). The compound showed identical NMR signals (Appendix A) to those described in the literature [26,27].

## 3. Materials and Methods

### 3.1. Materials

The standard ginsenosides Rb_1_, Rb_2_, Rb_3_, Rc, Rd, F_2_, R_7_, and Rd_2_ and the notoginsenosides Fa, Fc, Fe, Fd, and Gyp17 (HPLC ≥ 98%) were purchased from Vicky Biotechnology Co., Ltd. (Sichuan, China). *Panax notoginseng* flower was collected from Wenshan County, Yuannan Province, China, in October 2019 and was identified by a researcher of Xiuming Cui, Kunming University of Science and Technology (voucher No. YXY20191012). The Welchrom C_18_ column (4.6 × 250 mm, 5 μm) was purchased from Yuexu Technology Co., Ltd. (Sichuan, China). The Agilent 1260 High Performance liquid chromatograph was purchased from Agilent (Grand Island, NY, USA). The HSGF-_254_ silica gel plate was purchased from Yantai Jiang you Silica gel Development Co., Ltd. (Shandong, China). The Agilent 6530 Accurate-Mass Q-TOF LC/MS was from Agilent (Grand Island, NY, USA). The DEAE-52 was purchased from Shanghai Yuan ye Biological Co., Ltd. (Shanghai, China). *C. xylophilum* and other strains were isolated from *panax notoginseng* soil.

### 3.2. Isolation, Screening, and Species Identification of Fungi

Sixteen strains of fungi were isolated by the soil dilution plate method [28]. The isolated and purified strains were cultured on PDA medium, cultured at 26 °C for 3-4 days. The purified strain was stored in a refrigerator at 4 °C for subsequent studies. The strains with high ginsenoside transformation activity were screened through the transformation activity of ginsenoside Rb_1_. The amplification and sequencing of the ITS rDNA gene was completed by the Kunming Branch of Tsingke Biotechnology Co., Ltd (Branch of Tsingke Biotechnology Co., Ltd., Kunming, China). The isolated strain S7 was identified through morphological observation, biochemical characteristics, and phylogenetic analysis.

### 3.3. Preparation of Saponins in PNF

In this experiment, the PNF were extracted by the ethanol reflux extraction method. Sixty percent ethanol was used as the extraction solution; the liquid–solid ratio was 1:14; and the water bath at 60 °C was refluxed for 1.5 h, twice. The final extract yield was 40%. The extract was treated with D101 macroporous adsorption resin.

### 3.4. Biotransformation of Saponins in PNF by C. xylophilum

The biotransformation procedure was performed using PDB medium with 0.4 mg/mL saponins in PNF in a shaking incubator (160 rpm) at 26 °C for 15 days. Samples were withdrawn at regular intervals during fermentation (1, 5, 7, 10, 13, 15 d).

### 3.5. Biotransformation of Monomer Ginsenosides Rb_1_, Rb_2_, Rb_3_, Rc and Notoginsenosides Fa and Fc by C. xylophilum

The biotransformation procedure was performed using PDB medium with 0.05 mg/mL of the ginsenosides Rb_1_, Rb_2_, Rb_3_, and Rc and the notoginsenosides Fa and Fc in a shaking incubator (160 rpm) at 26 °C for 10 days.

### 3.6. Preparation and Purification of Crude Enzyme from C. xylophilum

#### 3.6.1. Preparation of Crude Enzyme

The culture medium was filtered with four layers of gauze to remove mycelia, and the supernatant was collected; When ammonium sulfate was added into the supernatant and the saturation reached 75%, the supernatant was precipitated for 1 h, then centrifuged (4000 r/min) for 20 min; the supernatant was discarded and the precipitation dissolved in HAc-NaAc (pH 5.0) buffer. The solution was centrifuged again (4000 r/min) for 20 min to remove the insoluble hybrid proteins. The crude enzyme solution was freeze-dried after dialysis for 24 h in HAc-NaAc buffer (pH 5.0).

#### 3.6.2. Purification of Crude Enzyme

The crude enzyme was purified by anion exchange column DEAE cellulose DE-52 (*ϕ*1.5 cm × 15 cm). The enzymatic activity of hydrolyzed ginsenoside Rb_1_ was detected, and the part of the hydrolyzed ginsenoside Rb_1_ was collected and then lyophilized. The purified protein was determined by Polyacrylamide gel electrophoresis (SDS-PAGE, Beyotime biotechnology, Shanghai, China).

### 3.7. Activity Analysis of Crude Enzyme from C. xylophilum

#### 3.7.1. *β*-glucosidase Activity Determination

Using pNP-*β*-*D*-glucopyranoside (pNPG) as a substrate, the activity of *β*-glucosidase was detected by colorimetry. The activity unit of *β*-glucosidase was defined as the amount of enzyme required for the hydrolysis of 1 mL enzyme solution for 1 min to produce 1 μmol *p*-nitrophenol (pNP).

#### 3.7.2. Biotransformation of Monomer Saponins by Crude Enzymes

The biotransformation procedure was performed as follows: dissolve the monomer saponins (ginsenosides Rb_1_, Rb_2_, Rb_3_, and Rc and notoginsenosides Fa and Fc) in 1 mL of pH HAc-NaAc buffer (pH 5.0) and mix with the same volume of crude enzyme; incubate at 50 °C for 2 days (the final substrate concentrations of the monomer ginsenosides were 0.05 mg/mL). In addition, the biotransformation of the monomer saponins by crude enzymes was studied in the pH range of 4 to 8 and the temperature range of 30 to 70 °C.

### 3.8. Preparation of Notoginsenoside Fe, Ginsenoside Rd_2_, and Notoginsenoside Fd from Main Saponins in PNF Transformed by C. xylophilum

The biotransformation procedure was performed using PDB medium with 0.4 mg/mL of saponins in PNF in a shaking incubator (160 rpm) at 26 °C for 15 days. The main saponins in PNF were transformed into minor ginsenosides by *C. xylophilum*. The cultivation of liquid was extracted with n-butanol 3 times, and the extract was concentrated under reduced pressure to obtain 21 g residue. The extract was eluted by D101 macroporous resin column chromatography with a gradient elution of an ethanol-water solvent system to obtain four fractions Fr. A~D. Fr. B was separated by repeated silica gel column chromatography (CH_2_Cl_2_-MeOH, 10:1~6:1) to obtain compound **1** (13.3 mg). Fr. C was separated by repeated silica gel column chromatography (CH_2_Cl_2_-MeOH, 10:1~5:1) to obtain compound **2** (35 mg) and compound **3** (17 mg).

### 3.9. General Analytical Methods

#### 3.9.1. Thin Layer Chromatography (TLC) Analysis

The thin layer chromatography (TLC) was performed using HSGF_254_ silica gel plates (Yantai Jiang you Silica gel Development Co., Ltd, Beijing, China) with CHCl_3_-CH_3_OH-H_2_O (6.3:6:0.2, *v*/*v*/*v*) as the developing solvent. The spots on the TLC plates were identified through comparisons with standard ginsenoside after visualization was made by spraying 10% (*v*/*v*) H_2_SO_4_ (in ethanol), followed by heating at 110 °C for 2 min.

#### 3.9.2. High-Performance Liquid Chromatography (HPLC) Analysis

HPLC analysis was performed using Welchrom C_18_ columns (4.6 × 250 mm, ID 5 µm; (Yuexu Technology Co., Ltd, Shanghai, China) connected to an Agilent 1260 HPLC system (NY, USA). The mobile phase consisted of water (A) and acetonitrile (B). The gradient elution was programmed as follows: 0–30 min, 20% (B); 30–60 min, 20–45% (B); 60–78 min, 45–75% (B); and 78–85 min, 75–100% (B).The flow rate was 1.0 mL/min, and the samples were detected by absorption at 203 nm. The injection volume was 20 µL. The column temperature was 30 °C.

## 4. Conclusions

This was the first report of the unique saponin conversion activities of *C. xylophilum*. Our study suggests that this fungus can convert the main saponins in the PNF to minor ginsenosides. When the monomer saponin is used as the transformation substrate, the transformation rate is high, and the transformation product is specific. Therefore, the fungus can specifically transform the main saponins in the PNF to produce minor ginsenosides, with a single transformation product and few by-products.

When the biotransformation of saponins in PNF (mainly including: ginsenosides Rb_1_, Rb_2_, Rb_3_, and Rc and notoginsenosides Fa and Fc) by *C. xylophilum*, the content of Fc was significantly reduced. However, when there was the biotransformation of the monomer notoginsenoside Fc by *C. xylophilum*, the Fc was not transformed. It was speculated that a promotion effect was produced between the saponins during the transformation of the main saponins in PNF by *C. xylophilum*. When Gpy17 was produced in the product, the transformation effect of notoginsenoside Fc was more obvious (Appendix A). This conjecture mainly refers to the research in this literature [29], and the combination of different substrates can be used for selective biotransformation.

We found that *C. xylophilum* isolated from *P. notoginseng* soil was highly effective and selective in the biotransformation of the main saponin (the notoginsenosides Fa and Fc and the ginsenosides Rb_1_, Rb_2_, Rc, and Rb_3_) in the PNF into minor saponins. The conversion rate was 100%, except for ginsenoside Rc at 88.5% and notoginsenoside Fc at 55.3%. The results of the present study suggest that *C. xylophilum* can be used to produce valuable minor ginsenosides from the main saponin in the PNF, with high biotransformation efficiency. These findings will lay a solid foundation for the construction of genetically engineered strains and eventually the large-scale preparation of minor saponins.

## Figures and Tables

**Figure 1 molecules-27-06615-f001:**
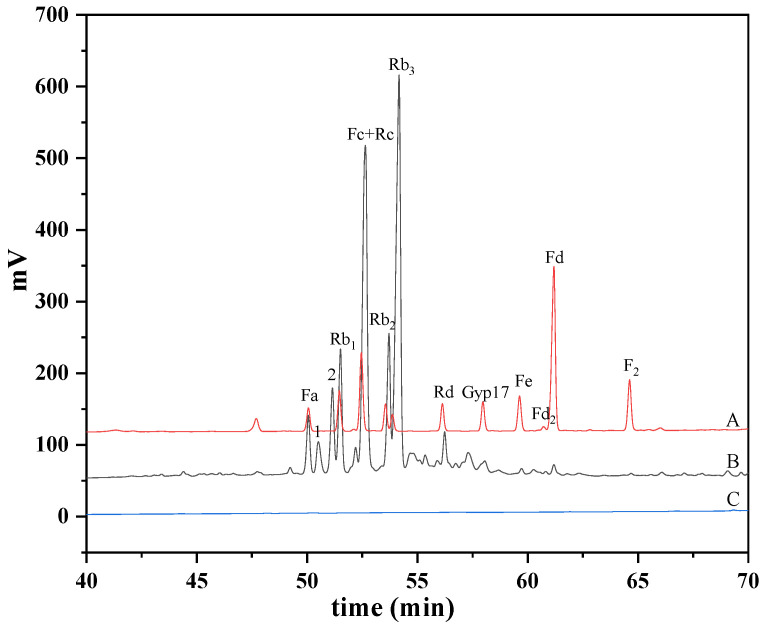
HPLC analysis of major saponins in PNF. (A) Commercial standards; (B) PNF extract; (C) control, culture liquid of strain S7 without added substrate; 1 and 2, unknown saponins.

**Figure 2 molecules-27-06615-f002:**
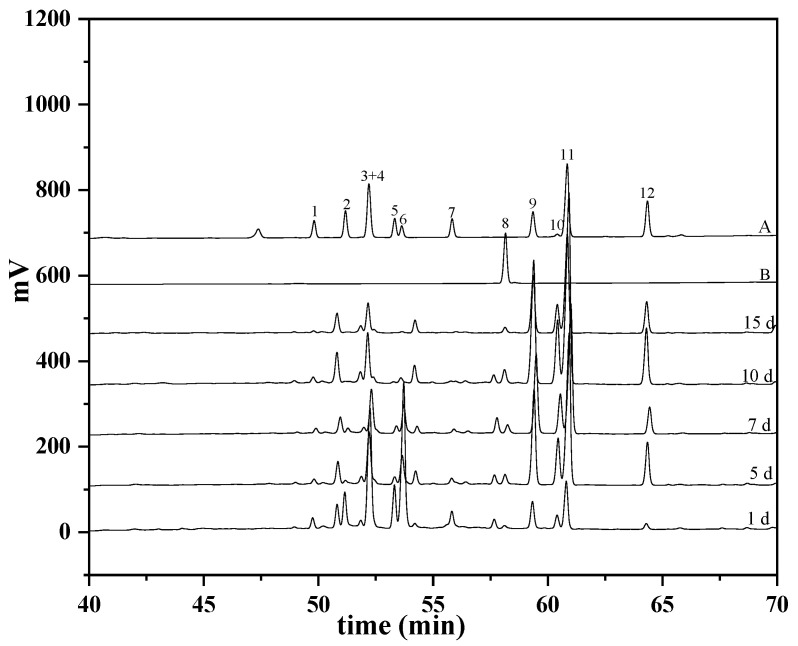
HPLC analysis of dynamic change of major saponins in PNF during biotransformation process by *C. xylophilum*. Twelve authentic saponins (A, B). The peaks: notoginsenoside Fa (1); ginsenoside Rb_1_ (2); notoginsenoside Fc (3); ginsenoside Rc (4); ginsenoside Rb_2_ (5); ginsenoside Rb_3_ (6); ginsenoside Rd (7); Gpy17 (8); notoginsenoside Fe (9); ginsenoside Rd_2_ (10); notoginsenoside Fd (11); ginsenoside F_2_ (12). Major saponins in PNF transformated by *C. xylophilum* for different days.

**Figure 3 molecules-27-06615-f003:**
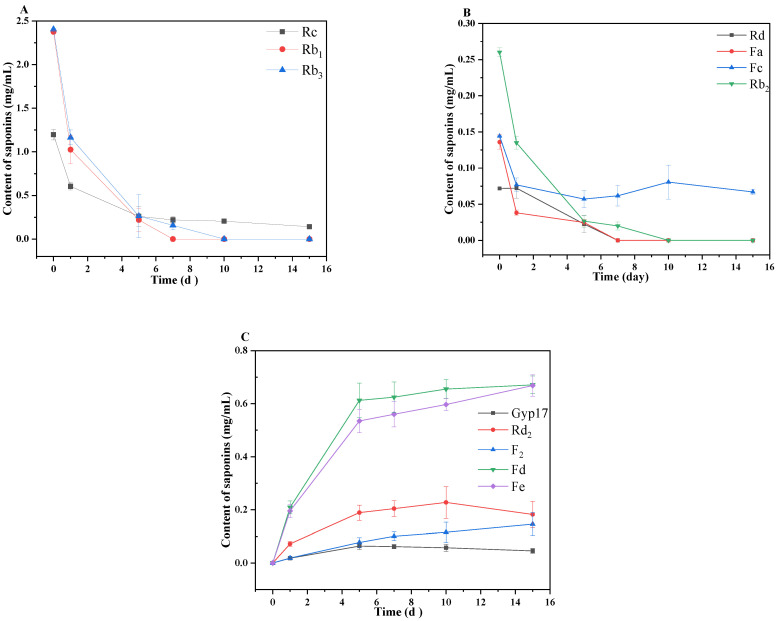
Dynamic change of substracts and products ginsenoside contents during biotransformation process by *C. xylophilum*. (**A**) Substracts (monomer ginsenosides Rc, Rb_1_, Rb_3_) contents; (**B**) substracts (monomer ginsenosides Rd, Rb_2_, notoginsenosides Fa, Fc) contents; (**C**) converted products of ginsenoside F_2_, Rd_2_, notoginsenoside Fd, Fe, and Gyp 17.

**Figure 4 molecules-27-06615-f004:**
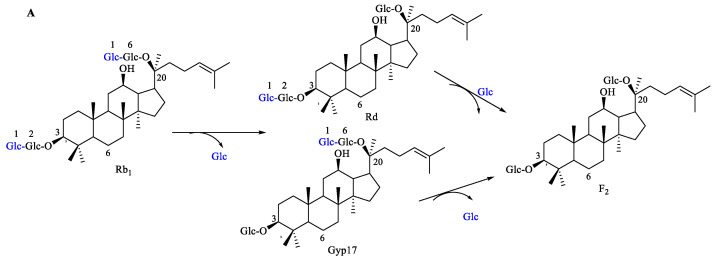
The proposed biotransformation pathways (**A**–**E**) of ginsenosides Rb_1_, Rb_2_, Rb_3_, and Rc and notoginsenoside Fa by *C. xylophilum*.

**Figure 5 molecules-27-06615-f005:**
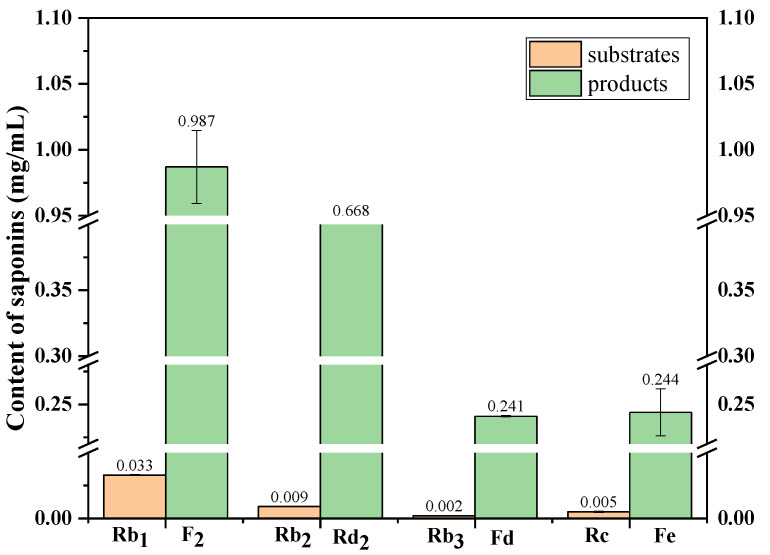
The contents change of ginsenosides F_2_ and Rd_2_ and notoginsenosides Fd and Fe from the transformation products of ginsenosides Rb_1_, Rb_2_, Rb_3_, and Rc by *C. xylophilum*.

**Figure 6 molecules-27-06615-f006:**
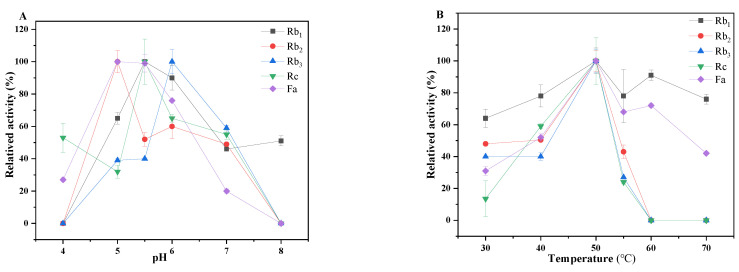
Effect on reactions: pH (**A**) and temperature (**B**) on the biotransformation of ginsenosides by crude enzymes.

## Data Availability

The data that support the findings of this study are available from the corresponding authors upon reasonable request.

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
