# Peer review of "Production of Minor Ginsenosides from Panax notoginseng Flowers by Cladosporium xylophilum"

_molecules, 2022, doi:10.3390/molecules27196615_

Round 1

Reviewer 1 Report

1. This is a very rigorous manuscript. The author first screened S7 from soil and identified it as Cladosporium xylophilum using ITS rDNA. Then the author explored the microbial transformation of C. xylophilum for Rb1 and so on. 2. However, some problems in the manuscript must be clarified and supplemented. A. p2 Figure 1. S17, S3.....should indicate which strain B. P3 line 95 Please explain in detail how 2.80, 0.29....% are obtained? C. How does P5 line 147 know ginsenoside R7 from MS experiments (Figure S2)? D. P8 Please supplement the 1H NMR spectra of compounds 1-3, and please indicate the D-solvent used. E. P8 line 228 HR-ESI MS is the correct representation F. P10 line 295297 Please specify volume of culture

Author Response

We have carefully revised our manuscript (ID: molecules-1894187),Please find our point-by-point reply to the reviewer’s comments. All modified places are marked in red. Our reply to review please see attachment. Thank you very much.

Reviewer 2 Report

The manuscript “Production of Minor Ginsenosides from Panax notoginseng Flowers by Cladosporium xylophilum” was submitted to Molecules for publication.

Broad comments:

The study describes cleavage of sugar moieties from the polyglycosylated major ginsenosides by a fungus detected in the soil of the tree. Thus, minor ginsensoides should be obtained which have proven stronger bioactivities due lower hydrophilicity.

Though the idea of the study is good and a lot of different experiments have been conducted, it cannot be published in its present form.

One reason is because it is very hard to follow which in some parts is due to language issues, but also because of the insufficient explanation of most of the experiments/figures.

One such example is the start of the study and the identification of a useful strain from the soil, which in my opinion is one of the highlights of the study. However, the reader does not really get this point until the materials and methods part and thus until the end of the manuscript. Instead, this should be discussed in the first part of the results and discussion, thereby also explaining why exactly S7 was chosen. Was it only because of one conversion product? Because S2, S3, and S17 also showed less polar spots on the TLC.

Also in section 2.4 the authors write “Based on the results obtained by HPLC analysis, …” without explaining what the reader should see. Apart from that, Figure 5 is cited after Figure 6A in the text, which should be avoided.

The phenomenon of insufficient information is coupled with unlucky graphical information, i.e.

figure legend 2: Please write commercial standards for A, and PNF extract for B;

figure legend 3: Please write compound names and number in parenthesis behind the name. Don’t summarize as done for e.g. 5-7. Days must not be explained, they are self-descriptive.

Figure 7. Please merge figures 7A and 7B and show all compounds with two bars (one before and one after digestion) for each compound. Thus, it will be self-informative.

Figures 1, 5, and 6 can be moved to the supporting information, as they are not relevant for the understanding of the manuscript.

The last major point, is that the authors suggest a bit more than actually happens. When talking of transformation, one expects e.g. new ring closures, formation of aromatic elements, etc. Instead, all that happens is deglucosylation in positions 3 and 20. Here, it has to be made clearer, what is the real benefit of the detected fungus or enzyme, respectively, in contrast to other available glucosidases.

Summarizing, the present manuscript would profit from language revisions of a native speaker as well as from a scientist not involved in this study, in order to increase the understandability of the written work. Maybe an overview of relevant ginsenosides for this study would be of help.

Specific comments:

Line 48:                              Please write “Biotransformation uses…”.

Line 53:                              If there are more advantages, please explain instead of writing “and so on”.

Lines 59:                            The authors cite two references in line 58, before writing “in this study” in line 59. Which of the two cited studies do they mean? Or do they mean the present study?

Line 60:                              Also here. Which study is “this study”.

Line 133:                           Please write C. Xylophilum in italics and Xylophilum in small letters. Especially the latter mistake occurs over and over again.

Author Response

We have carefully revised our manuscript (ID: molecules-1894187),Please find our point-by-point reply to the reviewer’s comments. All modified places are marked in red. Our reply to review please see attachment.

Round 2

Reviewer 2 Report

All suggestions were implemented.

Author Response

Reviewer(s)' Comments to Author:

Reviewer: 2

Thank you for making the suggested changes of the referees. I have reviewed your article and suggest one further change. I believe the article would be much more accessible to readers if the figure S3 from the SI data file is placed in main manuscript as its own figure - most likely in section 2.4. This will require you to change the figure numbers after the addition. With this change I would happy to accept your article for publication.

  Thank you for your kind suggestion, and we have moved Figure S3 from the SI data file into main manuscript in section 2.4 (Revised to Figure 4), please check.
